# Interleukin-33 Expression on Treatment Outcomes and Prognosis in Brazilian Breast Cancer Patients Undergoing Neoadjuvant Chemotherapy

**DOI:** 10.3390/ijms242216326

**Published:** 2023-11-15

**Authors:** Renata B. Albuquerque, Maria Amélia S. M. Borba, Matheus S. S. Fernandes, Tayrine O. Filgueira, Danyelly Bruneska G. Martins, José Luiz L. Filho, Angela Castoldi, Fabrício Oliveira Souto

**Affiliations:** 1Keizo Asami Institute (iLIKA), Federal University of Pernambuco (UFPE), Av. Prof. Moraes Rego s/n, Recife 50670-901, PE, Brazil; mborba@prospecmol.org (M.A.S.M.B.); tayrineordonioufpe@gmail.com (T.O.F.); bruneska@prospecmol.org (D.B.G.M.); joseluiz60@gmail.com (J.L.L.F.); castoldi.albuquerque@ufpe.br (A.C.); 2Postgraduate Program in Biology Applied to Health, Federal University of Pernambuco (UFPE), Av. Prof. Moraes Rego s/n, Recife 50670-901, PE, Brazil; 3Postgraduate Program in Neuropsychiatry and Behavioral Sciences, Federal University of Pernambuco (UFPE), Av. Prof. Moraes Rego s/n, Recife 50670-901, PE, Brazil; matheus.sfernandes@ufpe.br; 4Life Sciences Nucleus, Academic Center, Federal University of Pernambuco (UFPE), Rodovia BR-104, Km 59, s/n, Caruaru 55002-970, PE, Brazil

**Keywords:** interleukin-33, breast cancer, luminal subtype

## Abstract

Interleukin-33 (IL-33), a member of the interleukin-1(IL-1) family of cytokines, remains poorly understood in the context of human breast cancer and its impact on treatment outcomes. This study aimed to elucidate IL-33 expression patterns within tumor samples from a cohort of Brazilian female breast cancer patients undergoing neoadjuvant chemotherapy while exploring its correlation with clinicopathological markers. In total, 68 samples were meticulously evaluated, with IL-33 expression quantified through a quantitative polymerase chain reaction. The findings revealed a substantial upregulation of IL-33 expression in breast cancer patient samples, specifically within the Triple-negative and Luminal A and B subtypes, when compared to controls (healthy breast tissues). Notably, the Luminal B subtype displayed a marked elevation in IL-33 expression relative to the Luminal A subtype (*p* < 0.05). Moreover, a progressive surge in IL-33 expression was discerned among Luminal subtype patients with TNM 4 staging criteria, further underscoring its significance (*p* < 0.005). Furthermore, chemotherapy-naïve patients of Luminal A and B subtypes exhibited heightened IL-33 expression (*p* < 0.05). Collectively, our findings propose that chemotherapy could potentially mitigate tumor aggressiveness by suppressing IL-33 expression in breast cancer, thus warranting consideration as a prognostic marker for gauging chemotherapy response and predicting disease progression in Luminal subtype patients. This study not only sheds light on the intricate roles of IL-33 in breast cancer but also offers valuable insights for future IL-33-related research endeavors within this context.

## 1. Introduction

Breast cancer (BC) affects approximately 2.3 million women worldwide each year and stands as the leading cause of cancer-related deaths in over 100 countries [1]. The heterogeneity exhibited by clinical and pathological markers employed for classification, treatment, and prognosis underscores the necessity for personalized approaches to enhance patient outcomes [2,3].

Tumor classification hinges on the immunohistochemical profile, evaluating the expression of hormonal estrogen receptors (ER) and progesterone receptors (PR), human epidermal growth factor receptor 2 (HER-2), and Ki67 [2,4]. Consequently, breast tumors are categorized into four subtypes: Luminal A, Luminal B, HER-2 overexpressed, and Triple Negative (TNBC) [5,6]. Furthermore, these markers guide treatment protocols and prognostic assessment alongside the Tumor-Node-Metastases (TNM) staging system. Notably, histological subtypes classified as ER and HER-2-negative “good” types receive less aggressive treatment than TNBC, with systemic therapies reserved for cases of lymph node-positive disease [7]. In addition, the search for markers is increasingly necessary to define the development of cancer and also to be able to serve as a prognostic marker.

Interleukin-33 (IL-33), a cytokine within the (interleukin-1) IL-1 family, functions both as a nuclear factor and a classical cytokine, serving as prototypic alarmins [8,9]. These signaling pathways operate via growth stimulation expressed gene 2 protein (ST2) receptors, regulating inflammatory gene expression through the factor nuclear kappa B (NF-κB) and Mitogen-Activated Protein Kinase (MAPK) pathways [10,11]. The primary immunological agents within tumor tissue are Tumor-Infiltrating Lymphocytes (TILs), prominently present in breast cancer and representing a hallmark of this cancer type [12]. Their impact can either inhibit or contribute to tumor development, contingent upon the immune cell type and/or cytokines present [13]. In this context, IL-33 is increasingly recognized for its pivotal role in modulating innate and adaptive immunity, governing tissue homeostasis, and various pathological conditions, including allergy, infection, and cancer [14]. However, the precise role of IL-33 in breast cancer needs further investigation.

Nonetheless, studies have indicated that elevated IL-33 expression is linked to poor survival in gastric cancer, lung cancer, and hepatocellular carcinoma [15]. Moreover, IL-33 has been implicated in various cancer hallmarks such as oncogenesis, tumor growth, metastasis, neo-angiogenesis, and evasion of programmed cell death, also influencing the immune response against tumors [16]. In the context of breast cancer patients, elevated IL-33 levels are associated with an unfavorable prognosis [16,17]. Recent investigations have revealed that during breast cancer metastasis, IL-33 is upregulated in the lung microenvironment, primarily sourced from cancer-associated fibroblasts (CAFs) [18].

This study unveiled that stromal-derived IL-33 significantly contributes to type-2 inflammation and attracts diverse immune cell types to the lung microenvironment, ultimately fostering lung metastasis in breast cancer [18]. Although prior research has offered some insights, a comprehensive comprehension of the role and implications of clinical data pertaining to IL-33 expression in breast cancer remains elusive. Thus, this study investigated IL-33 expression in tumor samples from a cohort of Brazilian patients with breast cancer and correlated this expression with clinicopathological attributes and treatment options.

## 2. Results

Initially, we delineated the clinical and pathological characteristics of the studied sample of breast cancer, which are summarized in Table 1. The control group comprised six patients, with an average age of 36.8 ± 4.2 years. When we evaluated the molecular subtype of breast cancer, we noted that the patients were, on average, 54.76 years old at the time of diagnosis. The total number of chemotherapy- naïve patients was 52, while the number of patients who received neoadjuvant chemotherapy (NAC) was 16. The predominant histological classification was invasive breast carcinoma of no special type (IBC-NST), which represented 86.8% of the sample. According to immunohistochemistry, the most prevalent subtype was luminal B (50.0%). Most cases (44.1%) were diagnosed at late stages, with 58.8% showing lymph node involvement. The clinical tumor size was measured at 4.74 ± 0.41 cm (Table 1).

### 2.1. Chemotherapy-Naïve Breast Cancer Presents Increased IL-33 Expression

In order to determine if IL-33 expression is regulated in breast cancer, we aimed to assess whether there was a difference between the control group and the chemotherapy-naïve patient group. The comparative analysis of IL-33 mRNA expression in healthy breast and breast cancer samples demonstrated a significant up-regulation of IL-33 mRNA expression (Figure 1).

### 2.2. IL-33 Expression Is Upregulated in Luminal and TNBC Breast Cancer Molecular Subtypes

In order to elucidate whether there was a distinction in the expression of IL-33 among the breast cancer subtypes, we analyzed the expression of IL-33 in tumor samples stratified according to their breast cancer molecular subtype. Focusing on the group of chemotherapy-naïve patients, we observed an upregulation in all subtypes except for the HER-2-positive samples (Figure 2). Additionally, the expression of IL-33 was significantly higher in patients with the Luminal B subtype compared to those with the Luminal A subtype (*p* < 0.05). Notably, this upregulation was particularly pronounced in the subgroup of patients with triple-negative breast cancer (TNBC) (*p* ≤ 0.005).

### 2.3. IL-33 Expression Is Increased in the Luminal Subtype Independent of Lymph Node Involvement and TNM Stage

Considering the limited number of patients diagnosed with Her-2 overexpression (n = 6) and TNBC (n = 9) subtypes, we have chosen to extend this study exclusively, focusing on the luminal subtypes. Subsequently, we proceeded to assess whether the presence of axillary lymph node involvement had an impact on the expression of IL-33 mRNA. When analyzing lymph node involvement in the chemotherapy-naïve group compared to the control group, we observed an increase in IL-33 expression regardless of the presence or absence of lymph nodes. Additionally, we did not observe any correlation between the number of lymph nodes and IL-33 expression (Figure 3).

Aiming to investigate if breast cancer stage would be related to increased IL-33 expression, we employed a staging system based on the American Joint Committee on Cancer TNM criteria, which is widely used to guide the management of breast cancer. This classification system ranges from stage I to stage IV, with stage IV being recognized as the advanced or metastatic stage of the disease.

The analysis of IL-33 mRNA expression in breast cancer samples across TNM stages among Luminal subtypes in the chemotherapy-naïve group shows significant upregulation of IL-33 expression in subjects with all four cancer stages compared to the control group. This upregulation was particularly pronounced in stage IV. Additionally, a significant difference in IL-33 expression was observed between TNM stages I and IV (Figure 4).

### 2.4. Neoadjuvant Chemotherapy (NAC) Modulates IL-33 Expression in Luminal Breast Cancer Subtypes

Finally, we hypothesized the potential influence of chemotherapy on the expression of IL-33 in breast cancer tissue luminal subtypes. The clinical and pathological characteristics of the 16 patients who received neoadjuvant chemotherapy (NAC) are summarized in Table 1. The average tumor size for these patients before treatment was 5.5 ± 0.49 cm. Our findings revealed a significant decrease in the expression of IL-33 among both Luminal A (Figure 5A) and Luminal B (Figure 5B) patients who underwent NAC, in contrast to chemotherapy-naïve individuals. Moreover, when comparing the NAC group with the control group, no statistically significant differences were observed.

## 3. Discussion

Regarding the clinical and epidemiological aspects, the breast cancer patients enrolled in this study exhibit a pattern of disease distribution that mirrors that observed in Brazil and other parts of the world [1]. In contrast, diverging from the prevailing trend in developed nations, where breast cancer is often detected in its early stages, the majority of patients enrolled in this study were diagnosed with the disease at a more advanced stage. Regrettably, this observation underscores the particular practices employed by the public health system in the country [19].

Over 90% of the cases in this study were diagnosed with nonspecific-type invasive carcinoma, aligning with the findings reported in the literature. However, concerning the distribution of molecular subtypes among breast cancer patients, a higher proportion of 34 (50.0%) cases of luminal B subtype were observed, exceeding the reported range of 15–43% in the literature [20,21]. However, these values closely align with those found in recent studies conducted in Brazil, specifically in São Paulo [22], where Luminal A accounted for 29% and Luminal B for 47.5%, and in João Pessoa [23], with Luminal A at 22.5% and Luminal B at 57%.

Our study demonstrated that IL-33 mRNA was upregulated in chemotherapy naïve breast cancer samples when compared to the control group, thereby reinforcing its role as a tumor promoter in breast cancer. Previous studies in the Asian population have also reported upregulation of IL33 in breast cancer [24,25]. Interestingly, other studies have reported increased expression of IL-33 in other neoplasms, such as non-small cell lung cancer, liver cancer, squamous cell carcinoma, and head and neck cancer, when compared to control cases [15,26,27].

When the expression of IL-33 was analyzed in various subtypes of human breast cancer tissues, a significant increase in IL-33 expression was observed among patients with the Luminal B subtype compared to the Luminal A subtype. Conversely, downregulation of IL-33 expression was observed among patients with the HER-2 overexpression subtype compared to the control group.

The HER-2 receptor is a transmembrane tyrosine kinase receptor that belongs to the family of human epidermal growth factor receptors. The activation of HER-2 affects essential tumorigenic processes and plays a critical role in the pathogenesis of breast cancer, making it a key oncogene in this disease [28]. The increase in the number of HER-2 molecules activates several signaling pathways, including the MAPK (Mitogen-Activated Protein Kinase) and PI3K-AKT (Phosphatidylinositol 3-Kinase) pathways, which are involved in cell proliferation and survival [29]. The overexpression of HER-2 due to gene amplification occurs in 18% to 20% of breast cancers and is associated with a more aggressive phenotype. However, there are no studies in the literature that link IL-33 RNA expression to the HER-2 overexpression subtype.

Understanding the behavior of IL-33 in the tumor microenvironment requires a larger sample size, and the analysis of protein expression and cell frequency is necessary to elucidate the reason for its downregulation. Previous studies that evaluated IL-33 expression in breast cancer did not contrast between different subtypes. Thus, our data provides valuable information that can guide the development of new therapeutic options for specific subtypes.

Our analysis of different Luminal subtypes revealed an increase in IL-33 expression in the microenvironment associated with metastasis. The role of IL-33 in promoting or inhibiting tumor functions varies depending on the tumor environment [17]. In addition, prior studies have reported a significant upregulation of IL-33 expression in breast cancer patients with more than three lymph nodes involved through the use of immunohistochemical methods [24]. However, our study did not yield statistically significant differences between patients with lymph node involvement and those without in our comparisons.

Shani et al., demonstrated that IL-33 is upregulated in metastasis-associated fibroblasts in mouse models of spontaneous breast cancer metastasis and patients with breast cancer and lung metastasis [18]. Therefore, it is possible to infer that in breast cancer, IL-33 plays an immunosuppressive role that contributes to increased tumor aggressiveness. Although there are not many reports on the IL-33/ST2 pathway in breast cancer, Jovanovic et al. demonstrated in an animal model that mice lacking the ST2 receptor (ST2-deficient mice) and inoculated with the carcinogenic lineage (4T1) showed a decrease in tumor cell proliferation [16].

Other research findings indicate that IL-33 increases the level of amphiregulin-producing ST2+ Tregs in the lungs, which may promote metastatic tumor growth and induce a variety of biological effects, including cell proliferation, survival, migration, invasion, angiogenesis, and resistance to apoptosis [30,31]. We also observed a significant difference in the luminal subtypes of breast cancer that were detected in our study. This result is in agreement with prior investigations linking the Luminal B subtype to unfavorable clinic-histologic parameters and nodal metastasis [32].

Elevations in serum levels and immunohistochemical expression could promote BC progression and metastasis, and the relationship between IL-33 and immunosuppression in the tumor microenvironment was observed by IL-33 promotion of the Th2 immune response and mobilization of Treg cells, increasing the production of IL-5 and IL-13, thus contributing to the immunosuppressive tumor microenvironment [27]. In a separate study, researchers noted a positive correlation between advanced stages of breast cancer and increased expression of IL-33 mRNA, suggesting that this molecule may be involved in the progression of the disease [16]. The evidence presented in our study further underscores the significant role of IL-33 in the tumor microenvironment and calls for further prospective research to fully elucidate its functions and potential therapeutic implications for each subtype of breast cancer.

Our investigation of IL-33 expression regarding chemotherapy regimens revealed that NAC reduces the expression of IL-33 compared to chemotherapy-naïve individuals. However, no differences were observed when comparing NAC versus the control group. This suggests that NAC can modulate the expression of IL-33 to levels similar to those in the control group. Similar results were found in studies of lung cancer [33] and endometrial cancer in the serum of patients [34]. Previous studies have suggested that NAC is likely to activate an antitumor immune response in neoplastic lesions where it is already present or induce a response in cases where it is small or absent [35]. Our study contributes to the existing literature, highlighting the potential implications of chemotherapy schemes on IL-33 expression and the immune response in breast cancer patients. These findings warrant further investigation in larger, prospective studies.

Several studies have shown that ideal NAC increases TCD8 + cell infiltration and reduces forkhead box P3 (FOXP3) expression in the tumor microenvironment, leading to the induction of the antitumor immune response [36]. Apetoh (2007) [37] reported that cytotoxic agents, including anthracyclines, oxaliplatin, and radiation therapy, induce cancer and immunologic cell death. The destruction of an individual’s cells initiates an alert signal for the response of specific tumor T cells. Interestingly, T-cell induction is highly heterogeneous and varies according to each person and the characteristics of the tumor [13,37]. In addition, the heightened IL-33 expression within stromal elements, including CAFs, establishes an immensely inhibitory environment for the immune response [38].

In both subtypes, there was a lower expression in patients who received NAC than in those who received adjuvant chemotherapy. An Asian study also observed significantly higher serum IL-33 levels in pre-chemotherapy patients than in post-chemotherapy patients [39]. Furthermore, IL-33 is already known to significantly contribute to the advancement and metastasis of the tumor, primarily through the attraction of immune-suppressive cells like tumor-associated macrophages (TAM), myeloid-derived suppressor cells (MDSC), and regulatory T cells (Treg) [39].

Thus, our results suggest the potential utility of evaluated IL-33 expression as an indicator of the response to chemotherapy in patients with the luminal subtype and suggest that chemotherapy may be a modulator of inflammation in the tumor microenvironment, possibly diminishing tumor potential aggressiveness and thus down-regulating IL-33. In addition, the role of IL-33 in breast cancer progression and treatment response highlights its potential as a therapeutic target and prognostic indicator. Taking this into consideration, we examined existing cancer repositories for IL-33 gene expression. We identified repositories in the EMBL [40] and canSAR [41] databases that store gene expression data related to breast cancer. However, none of the genetic repositories we had access to specifically reported breast cancer subtypes in relation to IL-33 RNA expression. Thus, our study presented a distinct approach to analyzing IL-33 expression in different breast cancer subtypes.

Nevertheless, it is crucial to acknowledge the possible restrictions of this study. Firstly, the patients and individuals in the control groups were not matched in terms of age, which could introduce bias in the data since the healthy controls were younger than the breast cancer patients.

## 4. Materials and Methods

### 4.1. Study Cohort and Clinical Sample Processing

Study cohort and clinical sample processing his analytical cross-sectional study included 68 patients with operable breast cancer treated at Barão de Lucena Hospital, Recife-PE, Brazil. The inclusion criteria for this study included histologically confirmed invasive breast cancer and tumor-node-metastasis (TNM) with clinical staging between I and III (localized disease); the exclusion criteria included past neurological disorders or autoimmune diseases.

The clinicopathological and relevant demographic data for each of the patients were documented prospectively in our main breast cancer database. Cancer staging was performed according to the 8th edition of the American Joint Committee on Cancer TNM criteria [42], and breast cancer molecular subtypes were defined using immunohistochemistry-based analysis of estrogen, progesterone, HER-2, and Ki-67 receptor staining [2,4], which were obtained from the hospital’s medical records.

Healthy breast tissue samples from six women who had undergone plastic surgery for cosmetic purposes were collected and used as the control samples. All tissue samples were obtained from surgical specimens, and a fragment up to 0.5 cm in thickness was submerged in five volumes of RNAlater (Qiagen, Germantown, MD, USA) at room temperature for 2 h (to allow the solution to thoroughly penetrate the tissue) and then stored at −80 °C for total RNA extraction.

### 4.2. Ethical Considerations

All the patients signed an informed consent form for the use of their breast cancer tissue samples for research purposes. Ethics Committee approval was obtained for both cohorts studied in this research, according to the Code of Ethics of the World Medical Association number 47869315.0.0000.5208, and for the healthy breast tissues, we used the Certificate of Ethical Appreciation Presentation (CAAE) number 35626514.5.0000.5208, and the review number 852.334 was carried out with the Declaration of Helsinki.

### 4.3. Tissue RNA Extraction and DNase Treatment

Tissue RNA extraction and DNase treatment RNA extraction was performed using TRIzols (Invitrogen, Life Technologies, Horsham, UK) on breast cancer tissues. Breast tumor tissues were carefully dissected from the surrounding fatty tissue before TRIzol treatment. TRIzols reagent (1 mL) was added to 50–100 mg of tissue; the tissues were then homogenized using a mechanical tissue homogenizer at 0 °C. RNA was treated with DNase using a commercially available kit (Horsham, Ambion, UK), according to the manufacturer’s recommendations; after RNA extraction, the RNA concentration was determined using a NanoDrop spectrophotometer (NanoDrop ND1000; Thermo Fisher Scientific, Waltham, MA, USA). RNA integrity analysis was performed using gel electrophoresis, and total RNA was stored at −80 °C.

### 4.4. cDNA Synthesis and Gene Expression

For the functional analysis of IL-33, complementary DNA (cDNA) was obtained from mRNA using the Quantinova Reverse Transcription Kit^®^ (Qiagen, USA), following the manufacturer’s guidelines. Purity and concentration were assessed using a NanoDrop^®^—2000 spectrophotometer. All the samples were tested twice. The reaction was designed for a 10 mL final volume of 100 ηg/μL cDNA and 3.5 μM primers IL-33 (Fw: 5′-AGGCCTTCACTG AAAACAGG-3′ and Rv: 5′-TACCAAAGGCAAAGCACTCC-3′) in a 5 mL SYBR^®^ Green PCR Kit (7500) (Qiagen, USA), according to the manufacturer’s guidelines. B-Actin (Fw: 5′-CCTGGCACCCAGCACAAT-3′ and Rv: 5′-GCCGATACACGGAGTACT-3′) was used as an endogenous control. Quantitative PCR was performed at 7500 Fast Real-Time PCR^®^ (Qiagen, USA) with the following setup: initial denaturation at 95 °C for 10 min, followed by 40 cycles of 95 °C for 15 s, 60 °C for 50 s, and 30 s at 72 °C. The melting curve was analyzed to determine the quality of the reaction.

### 4.5. Data Processing and Statistical Analysis

The cycle threshold (Ct) limit values of the evaluated genes were subtracted from the Ct values of the endogenous controls, resulting in the ΔCt value. Subsequently, the Ct was calculated by subtracting the ΔΔCq value of each sample from the difference between the means of the ΔCt values of the endogenous control and each analyzed gene. The normalization step was performed according to the 2^−ΔΔCt^ method.

Comparisons between parameters with a normal distribution were performed using the independent-samples Student’s *t*-test, while non-parametric data were analyzed using the Mann–Whitney test. The normality of the data were evaluated using the Shapiro–Wilk test, and variables are described as the means ± standard error values. Differences between groups were evaluated using a one-way analysis of variance. Statistical significance was set at *p* < 0.05. GraphPad Prism version 9.0 (GraphPad Software, La Jolla, CA, USA) was used for all statistical analyses.

## 5. Conclusions

This study sheds light on the intricate relationship between Interleukin-33 (IL-33) expression and breast cancer, particularly within the context of Brazilian patients undergoing neoadjuvant chemotherapy. The findings indicate a significant upregulation of IL-33 expression in breast cancer samples, notably in the Luminal B and Triple-negative subtypes. Moreover, this study identifies a progressive increase in IL-33 expression among Luminal subtype patients with metastatic stage 4 TNM criteria, emphasizing its potential significance as a prognostic marker. Additionally, this study suggests that neoadjuvant chemotherapy could modulate IL-33 expression, potentially influencing tumor aggressiveness.

These findings contribute to our understanding of IL-33’s role in breast cancer progression and treatment response, highlighting its potential as a therapeutic target and prognostic indicator. This study underscores the importance of personalized treatment approaches based on molecular subtypes and encourages further research to fully unravel the complex interactions between IL-33, the tumor microenvironment, and treatment outcomes. As breast cancer continues to be a global health challenge, these insights offer valuable directions for future research and clinical interventions in the pursuit of improved patient outcomes.

## Figures and Tables

**Figure 1 ijms-24-16326-f001:**
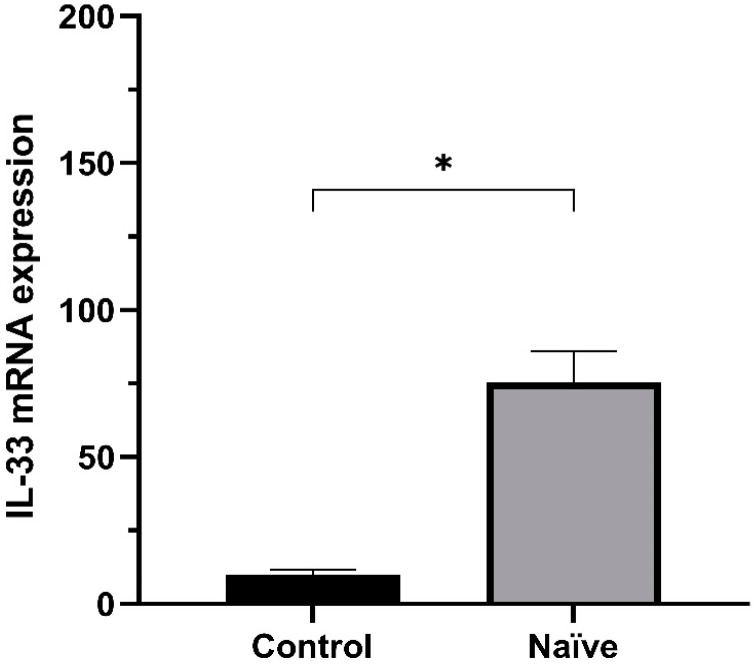
Differential IL-33 mRNA Expression across Chemotherapy-Naive Breast Cancer Subtypes and Healthy Samples. IL-33 mRNA expression using the threshold cycle (2^−ΔΔCT^), Mann–Whitney U test (* *p*  <  0.05). Naïve, n = 52; Control, n = 6. Naïve, Chemotherapy-naïve; Control, healthy breast tissues.

**Figure 2 ijms-24-16326-f002:**
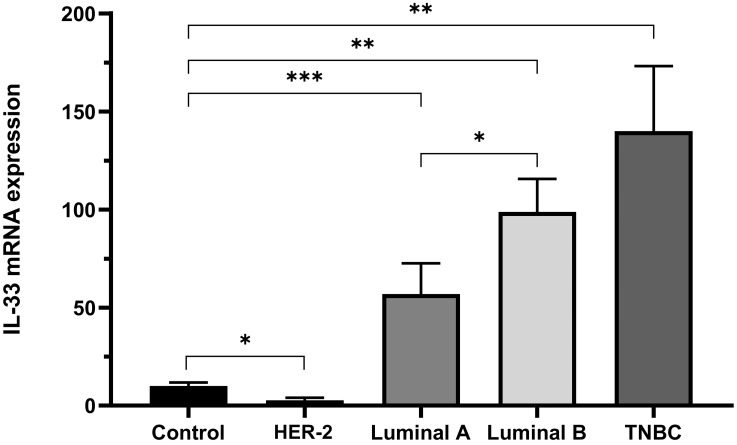
IL-33 Expression in breast cancer subtypes in chemotherapy-naïve patients. IL-33 mRNA expression using the threshold cycle (2^−ΔΔCT^), in the tissues of the healthy breast and breast cancer, was evaluated according to the profile that performed chemotherapy-naïve. Luminal A (LA), n = 14; Luminal B (LB), n = 25; Overexpression of HER-2 (HER-2), n = 6; and Triple Negative (TNBC), n = 7. Mann–Whitney U test (Control vs. HER-2; Luminal A; Luminal B; TNBC) and (Luminal A vs. Luminal B); (* *p* ≤ 0.05, ** *p* ≤ 0.005, *** *p* ≤ 0.001).

**Figure 3 ijms-24-16326-f003:**
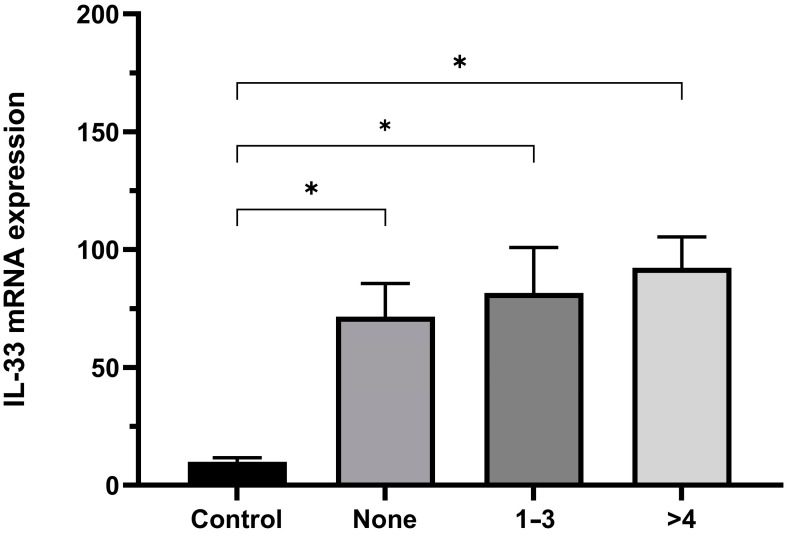
IL-33 Expression patient Luminal subtypes and number involvement of lymph nodes in the chemotherapy-naïve group. IL-33 mRNA expression using the threshold cycle (2^−ΔΔCT^). Mann–Whitney U test (control vs. none; 1–3; >4) (* *p* ≤ 0.05). None, n = 21; 1–3 lymph nodes, n = 16; and >4 lymph nodes, n = 2. None, absence of lymph node involvement; 1–3, 1–3 lymph node involvement; >4, >4 lymph node involvement.

**Figure 4 ijms-24-16326-f004:**
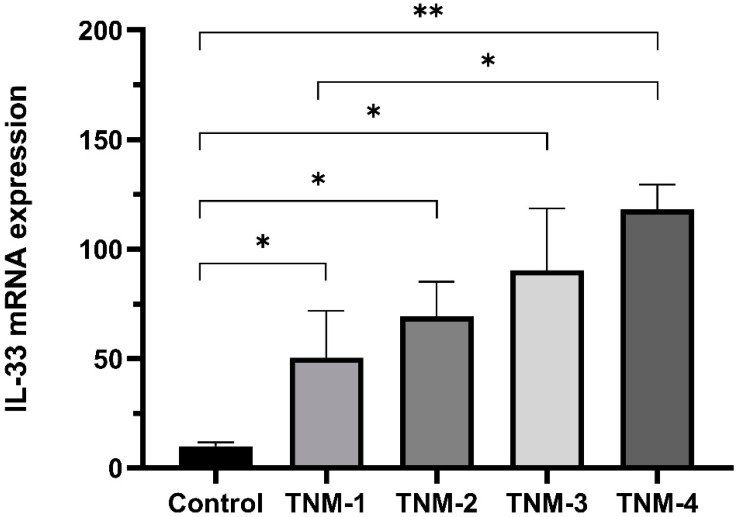
IL-33 mRNA expression in patients with Luminal subtypes and TNM staging criteria in the chemotherapy-naïve group. IL-33 mRNA expression using the threshold cycle (2^−ΔΔCT^). Mann–Whitney U test (Control vs. TNM-1; TNM-2; TNM-3) and (TNM-1 vs. TNM-4); (* *p* ≤ 0.05, ** *p* ≤ 0.005). Control (n = 6), TNM-1 (n = 8), TNM-2 (n = 20), TNM-3 (n = 6), and TNM-4 (n = 5).

**Figure 5 ijms-24-16326-f005:**
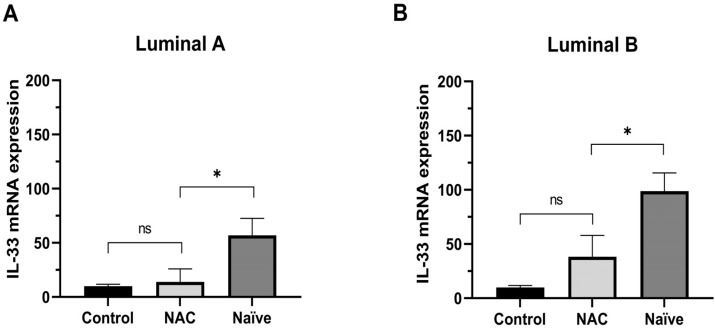
IL-33 mRNA expression in patients who underwent neoadjuvant and chemotherapy naïve compared to control in the Luminal A and B subtypes. (**A**) IL-33 mRNA expression in tissue breast cancer in the Luminal A that performed NAC (n = 5), Naïve (n = 14), and control (n = 6). (**B**) IL-33 mRNA expression in tissue breast cancer in the Luminal B that performed NAC (n = 9), Naïve (n = 25), and control (n = 6). IL-33 mRNA expression using the threshold cycle (2^−ΔΔCT^). Mann–Whitney U test (Control vs. NAC) and (NAC vs. Naïve) (* *p* < 0.05, ns = not significant). NAC, neoadjuvant chemotherapy; Naïve, chemotherapy-naïve.

**Table 1 ijms-24-16326-t001:** The clinical and pathological characteristics of this study participants.

Variables	All Patients (n = 68)	Chemo Naïve (n = 52)	NAC (n = 16)
Mean ± satandard			
Age (years-old)	54.76 ± 1.6	55.35 ± 1.8	52.88 ± 3.2
Clinical tumor size (cm)	4.74 ± 0.41	3.51 ± 0.39	5.5 ± 0.49
Frequency (n, %)			
Committed Lymph Nodes			
None	28 (41.2)	26 (50.0)	2(12.5)
1 to 3	32 (47.1)	24 (46.1)	8 (50.0)
≥4	8 (11.7)	2 (3.9)	6 (37.5)
TNM Staging			
I	9 (13.24)	9 (17.31)	0 (0.0)
II	28(41.18)	25 (48.08)	3 (18.75)
III	25 (36.75)	13 (25.00)	12 (75.00)
IV	6 (8.82)	5 (9.61)	1 (6.25)
Histologic type			
IBC-NST	59 (86.8)	46 (88.46)	13 (81.2)
ILC	2 (3.0)	1 (1.92)	1 (6.3)
Others	7 (10.2)	5 (9.62)	2 (12.5)
Molecular Subtype			
Luminal A	19 (27.94)	14 (26.9)	5 (31.3)
Luminal B	34 (50.0)	25 (48.1)	9 (56.2)
HER-2	6 (8.82)	6(11.5)	0 (0.0)
Triple-negative	9 (13.24)	7 (13.5)	2 (12.5)

IBC-NST, Invasive breast carcinoma of no special type; ILC, Invasive lobular carcinoma; HER-2, overexpression of HER-2; TNM, Tumor-node-metastasis breast cancer staging; NAC, Neoadjuvant chemotherapy.

## Data Availability

The original contributions presented in this study are included in the article. Further inquiries can be directed to the corresponding author/s.

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
