# Peer review of "Interleukin-33 Expression on Treatment Outcomes and Prognosis in Brazilian Breast Cancer Patients Undergoing Neoadjuvant Chemotherapy"

_ijms, 2023, doi:10.3390/ijms242216326_

Round 1

Reviewer 1 Report (New Reviewer)

Comments and Suggestions for Authors

Authors examine the expression of IL33 in Breast cancer patients undergoing Neo-adjuvant therapy. The study shows an increase in the expression of IL33 in luminal subtypes and TNBC, particularly advanced tumors. The Authors suggest Il33 as a prognostic marker. While the findings of the authors are important the authors must examine the existing cancer repositories for the expression of the gene. If the authors suggest this as a prognostic indicator disease outcome data from cancer repositories within the subtypes of the disease should be included.

Author Response

We appreciate your insightful comments on our study. Your suggestion to investigate existing cancer repositories for gene expression data related to breast cancer subtypes and IL-33 is highly valuable. In response to your feedback, we conducted a thorough search of repositories available within the EMBL-EBI and canSAR databases. While we did identify repositories containing gene expression data for breast cancer, none of them provided detailed information on IL-33 expression specific to different breast cancer subtypes.

In light of this, our study adopted a unique approach to examining IL-33 expression across various breast cancer subtypes, including luminal A, luminal B, HER2-positive, and triple-negative breast cancers. Our findings demonstrated significant variations in IL-33 expression within these subtypes. We want to extend our sincere gratitude for your input, as it has played a crucial role in enhancing the clarity and relevance of our research. Your valuable contribution has been incorporated into our paper, and we hope that our study will continue to contribute to the understanding of IL-33 as a potential prognostic marker in breast cancer.

Reviewer 2 Report (New Reviewer)

Comments and Suggestions for Authors

The finding that chemotherapy-naïve patients of Luminal A and B subtypes exhibited heightened IL-33 expression suggests a possible link between IL-33 and the aggressiveness of breast cancer tumors. The proposal that chemotherapy may mitigate tumor aggressiveness by suppressing IL-33 expression is an interesting hypothesis and could have important implications for breast cancer treatment strategies.

However, there are two critical points that require attention. Firstly, the authors should provide more information regarding the ethical considerations and procedures surrounding the use of healthy breast tissue controls. Transparency in this regard is essential for the ethical rigor of the study.

Secondly, it would be beneficial for the authors to discuss the potential use of immunohistochemistry (IHC) to validate the qPCR data. Combining qPCR with IHC can enhance the reliability and comprehensiveness of the findings by confirming the presence and localization of IL-33 in breast cancer tissue samples.

In conclusion, this study provides valuable insights into the complex roles of IL-33 in breast cancer and its potential impact on treatment outcomes. Addressing the ethical aspects of the study and considering the use of complementary techniques like IHC for data validation would strengthen the research and its implications for future IL-33-related investigations in the context of breast cancer.

Author Response

I want to express our gratitude for your insightful comments and questions. In response to your first comment, I would like to clarify that our study underwent a rigorous ethical review process. It was submitted to the Ethics Committee for Research Involving Human Subjects at the Federal University of Pernambuco, Center for Health Sciences. The study received a Certificate of Ethical Appreciation Presentation (CAAE) number 4786.9315.0.0000.5208, with a corresponding review number of 1.514.112. Additionally, for the collection of healthy breast tissue samples, we obtained a Certificate of Ethical Appreciation Presentation (CAAE) number 35626514.5.0000.5208, with a review number of 852.334. We ensured that all necessary documentation and procedures were in place to comply with ethical standards in human research. These details have been incorporated into the manuscript to enhance transparency.

Regarding your second comment, we appreciate your suggestion to include immunohistochemistry (IHC) as a complementary technique to validate our quantitative PCR (qPCR) data. We acknowledge that IHC can significantly improve the precision and reliability of our results by confirming the presence of IL-33 protein in breast cancer tissue samples. Unfortunately, we faced budget constraints due to the impact of the COVID-19 pandemic and previous government policies in Brazil, making it unfeasible for us to incorporate IHC into our study. Nevertheless, we believe that our findings, based on qPCR, still provide valuable insights into the expression of IL-33 in breast cancer subtypes and their potential implications for treatment outcomes.

We sincerely appreciate your feedback, as it has contributed to the overall quality and rigor of our research. Your thoughtful suggestions have been addressed to the best of our abilities, and we hope that our study will continue to be a valuable resource for future investigations into the role of IL-33 in breast cancer.

Round 2

Reviewer 1 Report (New Reviewer)

Comments and Suggestions for Authors

Thank you for the revisions.

Reviewer 2 Report (New Reviewer)

Comments and Suggestions for Authors

Accept in present form

This manuscript is a resubmission of an earlier submission. The following is a list of the peer review reports and author responses from that submission.

Round 1

Reviewer 1 Report

Comments and Suggestions for Authors

Albuquerque et al. present a study in which they identify IL-33 as a marker for advanced breast cancer (BC) in tissue samples of 68 Brazilian female patients. There is a detectable upregulation of IL-33 in the tissue of breast cancer patients compared to healthy control tissue with exception of Her2 driven breast cancer. IL-33 is also upregulated in metastatic patients as well as luminal B type compared to luminal A. In conclusion the provide some evidence for IL-33 as a tumor marker in the tumor tissues of Brazilian breast cancer patients with a chance to evaluate chemotherapy response and predicting disease progression in some patients.

Overall this is always an interesting subject, which has been also extensively researched in recent years and it is challenging to extract reasonable novelty from the data presented. There is a lack of care when it comes to description and precision in experimental readout that makes it a challenging read and leaves the reviewer more often than not questioning in which other ways the data could have been interpreted.

Several major concerns:

-      Supplemental data were not accessible

-      Expression data in all graphs are not identified (X-axis labelling is missing quantitative information). Is it fold increase?

-      It seems the authors used only parts of their cohort for some questions, patients with metastasis and no metastasis need to be clearly identified. Which ones were used in Figure 3?

-      Luminal A/B comparison in figure 4 is a duplication of figure 2 even with TNM staging. At least show also the differences for all the other stages.

-      Figure 5 again feels like a duplication of figure 1, why do they authors not show TNBC? 7/9 patients were chemo naive…?

Minor concerns:

-      Spelling mistakes and typos throughout the manuscript, please revise

-      Over-interpretation of data: ”Our findings suggest that chemotherapy may reduce tumor aggressiveness by decreasing IL-33 expression in breast cancer tissue ...” Where exactly is this tested functionally in this study? One can speculate about connections but that should be part of a discussion.

Comments on the Quality of English Language

-      Spelling mistakes and typos throughout the manuscript, please revise

Author Response

Dear reviewer, thank you for all your suggestions, questions, and comments. We made the changes suggested in the text and here we answer point by point the questions raised.

  1. Supplemental data were not accessible.

We apologize for the error, our manuscript does not have any supplementary material to add to the journal. Thanks for the comment.

  1. Expression data in all graphs are not identified (X-axis labelling is missing quantitative information). Is it fold increase?

Thanks for the contribution, we have made changes to all graphs, including the mRNA relative expression information, and added the method used in each caption.

  1. It seems the authors used only parts of their cohort for some questions, patients with metastasis and no metastasis need to be clearly identified. Which ones were used in Figure 3?

In response to your comment, we appreciate your insightful feedback. In our study, we acknowledge that only certain segments of our cohort were used for specific questions. To address this concern, we agree that it is crucial to clearly identify patients with metastasis and those without metastasis. We have made modifications to both the text and the graphical representation (Figure 5) to ensure better clarity regarding this information. We must emphasize that our study presents preliminary comparative data, and due to the small number of patients involved, there is a possibility of limited representativeness and potential type two systematic errors in our correlation analysis. However, we recognize the importance of conducting a more comprehensive analysis in future studies to address these limitations.

  1. Luminal A/B comparison in figure 4 is a duplication of figure 2 even with TNM staging. At least show also the differences for all the other stages. 

Thank you for your comment regarding the comparison of Luminal A/B subtypes in Figure 4 of our study. We want to assure you that we have carefully considered your comment and made the necessary modifications to improve clarity and avoid duplication. In Figure 2, we evaluated the control group alongside different subtypes of breast cancer. In Figure 4, we have made the suggested modification and clearly depicted all TNM stages. This figure now focuses specifically on the Luminal subtype and provides a comprehensive comparison. In addition to comparing the Luminal subtype with the control group, we have also identified and described differences within stage I, as stated in the manuscript. This allows for a more detailed analysis and interpretation of the data. We greatly appreciate your insightful comment, and your feedback has helped us enhance the clarity and comprehensiveness of our study. Thank you for bringing this to our attention, and we are committed to continuously improving the quality and accuracy of our research.

  1. Figure 5 again feels like a duplication of figure 1, why do they authors not show TNBC? 7/9 patients were chemo naive…?

Unfortunately, we have only 2 patients according to table 1 with neoadjuvant chemotherapy treatment, so it was not possible to carry out the analyzes for the TNBC subtype.

Minor concerns:

  1. Spelling mistakes and typos throughout the manuscript, please revise.

We adjust as requested. Thanks for your contribution.

  1. Over-interpretation of data: ”Our findings suggest that chemotherapy may reduce tumor aggressiveness by decreasing IL-33 expression in breast cancer tissue ...” Where exactly is this tested functionally in this study? One can speculate about connections but that should be part of a discussion.

We have taken this feedback into account and made the necessary revisions. We have included a more explicit explanation in the discussion section to clarify that the connection between chemotherapy, IL-33 expression, and tumor aggressiveness is speculative and requires further investigation. Additionally, as recommended by another reviewer, we have incorporated a comparison with the control group on the same chart to provide additional support for this hypothesis. To elaborate further, our research findings indicate that a potential reduction in IL-33 expression associated with chemotherapy could be observed by analyzing breast cancer tissue NAC levels in relation to patients receiving naïve chemotherapy. However, it is important to note that the values for tissue NAC do not exhibit significant differences among the control group. By making these adjustments, we aim to ensure transparency and accuracy in our study and to emphasize the need for further discussion and exploration of the potential connections between chemotherapy, IL-33 expression, and tumor aggressiveness. Thank you for bringing this concern to our attention, and we appreciate your valuable input in refining the interpretation and discussion of our data.

We added an explanation to the discussion to make this speculation clearer and added the comparison with the control group on the same chart as recommended by the other reviewer to make this hypothesis more substantiated.

The research findings suggest that chemotherapy may reduce IL-33 expression could be observed by decreasing breast cancer tissue NAC in relation to patients with naïve chemotherapy, and the values for tissue NAC do not represent significant differences among the controls.

Reviewer 2 Report

Comments and Suggestions for Authors

I. Review report to the Authors:

Summary / significance: This article by Albuquerque et al. focuses on the expression of IL-33 supposed to be associated with aggressiveness of breast cancer and regulation of immune response. Targeting IL-33 signaling holds promise for treatment of patients by dampening tumor immunosuppression. The study connects to previous work of the authors in which expression of miRNA21 was analyzed in exactly the same manner.

Experiments, data and conclusions: IL33 mRNA expression was evaluated by qRT-PCR in fresh breast cancer specimens and healthy controls. The authors test IL33 expression levels in different BrCa subtypes and after neoadjuvant chemotherapy proposing it a prognostic marker in BrCa and that therapy downregulates IL-33 expression and enhances anti-tumor immunity. The experimental approaches shown are in principle well described and executed accurately, and the results drawn seem to prove diverse IL33 levels.

There are major concerns:. Data description needs clarification and amendments: results have to be described coherent regarding the tumor subtypes and sub-groups.These have to be specified before showing comparisons of IL-33 levels, there is no coherence between the data which makes the whole interpretation questionable. In the Figures, inscriptions and y-axis scales have to be improved to clarify the contents

Publication of this article is not possible at this stage.

Major comments:

1) Results Table 1: an own subsection “Clinical characteristics of the BrCa cohort” has to be introduced.

How many patients were in the control group, you mention n=6 in the Fig.legend1? Tumor size 4.74 is without size unit. Did you correlate IL-33 levels with tumor size or any other parameter?

2) Results Headers 2.1 and 2.2: title “mRNA-33” is insufficient, better “IL33 mRNA expression…” (IL33 is the gene name, therefore more correct than the protein name IL-33).

3) Data descriptions generally: Tumor subtypes and groups have to be clearly described and specified before showing comparisons of IL-33 levels. Results in all subsections 2.1 – 2.3. are just listed without comment, introductory, and concluding sentences are missing.

4) Figures generally: The use of different y-axes scalings makes the quantitative comparison difficult or impossible. In all figures: what is the readout shown on y-axis? Why are control levels diverse in the different figures? There are too many abbreviations in the figures which makes the message hard to understand. Figure legends have to be extended and improved.

5) Figure 1: in the text should be mentioned that the chemo-naïve were n=52 and referred to Table 1.

6) Figure 2: again: y-axis inscription, how many patients in the control? Is the control level the same value as shown in Fig.1?

7) Table 2: is not understood. “BrCa prognostic factors” is wrong, it is the TNM description. What is the readout/measure of IL-33 mRNA shown? What means “difference compared to the control samples”? These are completey different numbers. Why not put this into a bar chart?

8) Figure 3: also this subgroups are unclearly described, what is meant with “between luminal subtype”? controls missing, what is the non-meastasis group? A combinatin of all tumors?

9) Figure 4: again controls missing. Ist the stage III the same as grade what is III-LA –LB? please spedify

10) Figure 5: If showing downregulation in NAC, this should be shown right of naïve, and the controls are essential to compare with baseline levels. Mention luminal A and luminal B above the panels. How come that in Fig.1 you show 52 naïve patients and in Figure 5 n=9 and n=25 naïve patients? What´s the difference in these naïve groups?

Minor comments:

1) Abstract: better shift sentences 3 and 4: first the number of patients in the cohort, then the type of analysis.

2) Abstract: the results should be described more precisely, finally it remains unclear which IL-33 increase was observed against which cancer types.

3)  Abstract: “The IL-33 is a tumor-promoting in the tumor tissues…” does not make sense.

Comments on the Quality of English Language

English level: The English phrasing needs thorough improvements. Please check it with a spelling program or have a native speaker read the manuscript.

Author Response

Dear reviewer, thank you for all your suggestions, questions, and comments. We made the changes suggested in the text and here we answer point by point the questions raised.

Major comments:

  1. Results Table 1: an own subsection “Clinical characteristics of the BrCa cohort” must be introduced. How many patients were in the control group, you mention n=6 in the Fig.legend1? Tumor size 4.74 is without size unit. Did you correlate IL-33 levels with tumor size or any other parameter?

Thank you for your question and feedback regarding Table 1 in our study. We have made the necessary clarifications based on your comment. In the control group, we apologize for any confusion caused. The control group comprised six healthy breast tissue samples. We have corrected this information in the text to accurately reflect the number of patients in the control group. Regarding the tumor size mentioned as 4.74, we apologize for the omission of the size unit. It should have been stated alongside the measurement. We understand the importance of providing clear and complete information, and we apologize for any confusion caused by this oversight. Regarding the correlation analysis between IL-33 levels and tumor size or other parameters, we acknowledge that our study consists of preliminary comparative data. Due to the small number of patients in our study, the representativeness of the correlation analysis could be limited. However, we are aware of the significance of conducting such an analysis, and we plan to explore these correlations in future studies. We appreciate your attention to these details and your valuable feedback. Your input helps us improve the accuracy and clarity of our study.

  1. Results Headers 2.1 and 2.2: title “mRNA-33” is insufficient, better “IL33 mRNA expression…” (IL33 is the gene name, therefore more correct than the protein name IL-33).

Thank you for your feedback regarding the Results Headers 2.1 and 2.2 in our study. We appreciate your attention to detail and the suggestion for improvement. The information has been added to the text as requested.

  1. Data descriptions generally: Tumor subtypes and groups have to be clearly described and specified before showing comparisons of IL-33 levels. Results in all subsections 2.1 – 2.3. are just listed without comment, introductory, and concluding sentences are missing. 

We appreciate your feedback and understand the importance of providing clear and specific descriptions of tumor subtypes and groups before presenting comparisons of IL-33 levels. Based on your suggestion, we have made the necessary revisions to address this concern, the information has been added to the text as requested.

  1. Figures generally: The use of different y-axes scaling’s makes the quantitative comparison difficult or impossible. In all figures: what is the readout shown on y-axis? Why are control levels diverse in the different figures? There are too many abbreviations in the figures which makes the message hard to understand. Figure legends have to be extended and improved. 

We appreciate your feedback and understand your concerns regarding the y-axis scaling. To address these issues, we will make adjustments to the y-axis formatting in the control group to improve visualization and facilitate easier comparison. We acknowledge the importance of consistency and clarity in presenting the control levels across different figures. Additionally, we have taken steps to enhance the figure legends by improving and expanding the descriptions. This will provide more comprehensive information and make the figures easier to understand, reducing confusion caused by excessive use of abbreviations.

  1. Figure 1: in the text should be mentioned that the chemo-naïve were n=52 and referred to Table 1. 

The information has been added to the text as requested.

  1. Figure 2: again: y-axis inscription, how many patients in the control? Is the control level the same value as shown in Fig.1?

Thank you for your question regarding Figure 2 in our study. We appreciate your attention to detail and apologize for any confusion caused. To address your query, the number of patients in the control group is indeed the same in both Figure 1 and Figure 2. The control group level depicted in Figure 2 corresponds to the same value shown in Figure 1. Furthermore, we acknowledge the need for clarity and better visualization of the y-axis inscription. We will make the necessary adjustments to the axis formatting to enhance readability and improve the overall presentation of the figure.

  1. Table 2: is not understood. “BrCa prognostic factors” is wrong, it is the TNM description. What is the readout/measure of IL-33 mRNA shown? What means “difference compared to the control samples”? These are completey different numbers. Why not put this into a bar chart?

Thank you for your comment and feedback regarding Table 2 in our study. We appreciate your observations and concerns about the understanding of the table and the presentation of the data. To address your points, we have made several improvements based on your excellent commentary. First, we acknowledge that the term "BrCa prognostic factors" was incorrect, and we apologize for any confusion caused. We have corrected it to accurately reflect the TNM description. Regarding the readout or measure of IL-33 mRNA shown in Table 2, we understand the need for clarity. We have provided additional information in the captions to specify the exact units or measurement values for IL-33 mRNA. Additionally, we have chosen to describe the histological grade result in text format to provide a clearer understanding. In the captions, we have elucidated the respective comparisons between the groups through the Mann-Whitney test, ensuring transparency and clarity in the statistical analysis.

  1. Figure 3: also this subgroups are unclearly described, what is meant with “between luminal subtype”? controls missing, what is the non-meastasis group? A combinatin of all tumors? 

We evaluated the expression of IL-33 mRNA in patients classified as Luminal based on immunohistochemical staining, which includes both Luminal A and Luminal B subtypes. Within this group, we compared the patients who presented with metastasis (TNM-IV) and those who did not. As suggested, we have included a comparison of these groups with the control group. We change the text and the graph can be more clear the results (figure 4). 

  1. Figure 4: again controls missing. Ist the stage III the same as grade what is III-LA –LB? please spedify.

First, I would like to express gratitude for your insightful comment. In this graph, we evaluated the TNM stage III for Luminal A (LA) and Luminal B (LB) subtypes and observed that Luminal A in stage III exhibited lower IL-33 expression compared to Luminal B in the same stage. However, upon revisiting the figure, we discovered an error in our analysis, as indicated in the legend. The analyzed sample size was n=15 because we inadvertently included patients who underwent neoadjuvant chemotherapy (NAC) and were chemotherapy-naïve, which was not the intended focus of our study. In Table 1, the total number of patients in stage 3 and chemotherapy-naïve was 13. The comparative analysis was intended to compare the chemotherapy-naïve group between Luminal A - stage III (n=3) and Luminal B - stage III (n=4). However, upon conducting this comparative analysis, we did not observe any significant difference. Therefore, we have decided to remove this information as it holds no relevance to the study.

  1. Figure 5: If showing downregulation in NAC, this should be shown right of naïve, and the controls are essential to compare with baseline levels. Mention luminal A and luminal B above the panels. How come that in Fig.1 you show 52 naïve patients and in Figure 5 n=9 and n=25 naïve patients? What´s the difference in these naïve groups? 

In Figure 1, all breast cancer subtypes that underwent chemotherapy Naïve are 52, while in Figure 5 only the analysis of Chemotherapy Naïve Luminal A (Figure 5A) and Chemotherapy Naïve Luminal  B (figure 5B) was demonstrated. Hence, the values differ between the two figures. These values are also described in Table 1, which provides a clearer stratification of the number of patients for subtypes. In the graph, we added the control group to enable a comparison with the baseline data.

Minor comments:

  1. Abstract: better shift sentences 3 and 4: first the number of patients in the cohort, then the type of analysis. 

We appreciate the contribution and change the sentence in the text.

  1. Abstract: the results should be described more precisely, finally it remains unclear which IL-33 increase was observed against which cancer types.

We appreciate the contribution and change the sentence in the text.

  1. Abstract: “The IL-33 is a tumor-promoting in the tumor tissues…” does not make sense.

We appreciate the contribution and change the sentence in the text.

Round 2

Reviewer 1 Report

Comments and Suggestions for Authors

In my opinion, the issues remain largely unadressed, even with the authors providing some additional datapoints. 

Comments on the Quality of English Language

acceptable

Reviewer 2 Report

Comments and Suggestions for Authors

The revised manuscript by Albuquerque et al. addresses all comments and concerns. The authors have improved the text and especially the figures, and they have rearranged and modified sections in a way that it is now clear to read.

Comments on the Quality of English Language

English still needs some rephrasing.